# Combating Bacterial Biofilms: Current and Emerging Antibiofilm Strategies for Treating Persistent Infections

**DOI:** 10.3390/antibiotics12061005

**Published:** 2023-06-03

**Authors:** Ahmed G. Abdelhamid, Ahmed E. Yousef

**Affiliations:** 1Department of Food Science and Technology, The Ohio State University, 2015 Fyffe Court, Columbus, OH 43210, USA; abdelhamid.9@osu.edu; 2Botany and Microbiology Department, Faculty of Science, Benha University, Benha 13518, Egypt; 3Department of Microbiology, The Ohio State University, 105 Biological Sciences Building, 484 West 12th Avenue, Columbus, OH 43210, USA

**Keywords:** bacterial biofilm, biofilm infection, antibiofilm agents, quorum sensing, antimicrobial peptides, antibiofilm nanoparticles

## Abstract

Biofilms are intricate multicellular structures created by microorganisms on living (biotic) or nonliving (abiotic) surfaces. Medically, biofilms often lead to persistent infections, increased antibiotic resistance, and recurrence of infections. In this review, we highlighted the clinical problem associated with biofilm infections and focused on current and emerging antibiofilm strategies. These strategies are often directed at disrupting quorum sensing, which is crucial for biofilm formation, preventing bacterial adhesion to surfaces, impeding bacterial aggregation in viscous mucus layers, degrading the extracellular polymeric matrix, and developing nanoparticle-based antimicrobial drug complexes which target persistent cells within the biofilm core. It is important to acknowledge, however, that the use of antibiofilm agents faces obstacles, such as limited effectiveness in vivo, potential cytotoxicity to host cells, and propensity to elicit resistance in targeted biofilm-forming microbes. Emerging next generation antibiofilm strategies, which rely on multipronged approaches, were highlighted, and these benefit from current advances in nanotechnology, synthetic biology, and antimicrobial drug discovery. The assessment of current antibiofilm mitigation approaches, as presented here, could guide future initiatives toward innovative antibiofilm therapeutic strategies. Enhancing the efficacy and specificity of some emerging antibiofilm strategies via careful investigations, under conditions that closely mimic biofilm characteristics within the human body, could bridge the gap between laboratory research and practical application.

## 1. Introduction

Bacterial biofilms are communities of cells that are attached to surfaces and to each other, and embedded in a self-produced matrix consisting of extracellular polysaccharides (exopolysaccharides), proteins, and extracellular DNA (eDNA) [1]. When biofilms are formed inside living organisms, imbedded bacteria are protected from the host defense system and their survival is enhanced by employing various strategies such as dormancy. Bacteria within biofilms exhibit altered gene expression and metabolism in response to environmental anoxia and nutrient limitation, which lead to a reduced metabolic rate and cell division [2,3]. These adaptations may confer antimicrobial resistance by inactivating the antimicrobial receptors on microbial cells or reducing the cellular functions that the antimicrobials interfere with. Biofilm infections trigger innate and acquired host immune responses that may not eliminate the biofilm pathogen, but instead, result in collateral tissue damage [4]. Biofilm-related diseases are chronic infections that develop slowly, cannot be resolved efficiently by the immune system, and respond inconsistently to antimicrobial treatments.

The current review assesses the health risks posed by biofilm infections, a historically emphasized yet currently pertinent area of concern. The article provides an in-depth analysis of contemporary knowledge and technological advancements in mitigation strategies designed to combat bacterial biofilms associated with infections. The review underscores the promise of emerging antibiofilm small molecules that not only mitigate bacterial biofilms but also alleviate the often-associated antimicrobial resistance. Furthermore, the review provides a detailed exploration of future research paths aimed at advancing antibiofilm therapy.

## 2. Role of Biofilms in Persistent Infections: An Overview

Serious clinical problems arise when a species or multiple species of pathogenic bacteria develop persistent infections, such as those associated with biofilm buildup (Figure 1). Persistent infections are difficult to treat with antimicrobials, and thus, the spread of infection ensues. Biofilm-associated infections can be classified into (i) surface-located biofilms, which are formed on biotic (e.g., gingiva) or abiotic (e.g., catheter) surfaces, and (ii) tissue- or secretion-located biofilm, where bacteria aggregate within infected tissues or accumulate in secretions (e.g., mucus). Surface-located biofilms are precursors of intravascular catheter infections, prosthetic joint infections, and chronic wounds such as diabetic foot ulcers [5]. A typical example of surface-located biofilms is dental biofilms which are complex multispecies structures found in the oral cavity and composed of aerobic and anaerobic microorganisms. The microbial genera involved in dental biofilms include *Streptococcus*, *Neisseria*, *Candida*, *Peptostreptococcus*, *Actinomyces*, *Veillonella*, and *Fusobacterium* [6]. Dental biofilms are responsible for causing caries and periodontitis. Tissue-located biofilms are associated with host predisposing factors such as genetic mutations or bone trauma, which create a favorable environment for the establishment and growth of biofilms [7]. Examples of such factors include a mutation in the gene coding the cystic fibrosis transmembrane regulator protein, which leads to the accumulation of viscous secretions in the lungs [8] and an increased risk of bacterial biofilm infections, or the formation of bone sequestrum after bone trauma which creates a site for bacterial infection [9]. These infections result in chronic inflammation and destruction of the affected tissue, such as lung tissue in cystic fibrosis, heart valves in infective endocarditis, or bones in nonhealing wounds. Generally, any microorganism capable of forming bacterial aggregates that confer biofilm characteristics can cause biofilm-related infections, with *Staphylococcus epidermidis*, *Staphylococcus aureus*, *Pseudomonas aeruginosa*, and certain members of *Enterobacteriaceae* being the most common. The dispersal of biofilms can lead to the colonization of new niches, leading to new biofilm infections. Staphylococci are the most common cause of infections associated with indwelling medical devices [10]; the associated biofilm infections, particularly those caused by highly virulent *S. aureus* strains, often result in severe local infection or sepsis. The most-described tissue-located biofilm infection is the one resulting from *P. aeruginosa* colonization in cystic fibrosis lungs.

## 3. Bacterial Biofilm Formation and Characteristics by Infection Site

Despite many common features, biofilm characteristics and functions vary by the location where it is formed. Surface and tissue locations will be addressed in this article.

### 3.1. Surface-Located Bacterial Biofilms

Biofilms are manifested as complex aggregates of microorganisms within extracellular polymeric substances (EPSs) and the resulting structures are irreversibly attached to surfaces. The formation of EPSs occurs in the attachment stage of a biofilm to the surface. The EPSs are composed mostly of polysaccharides, nucleic acids, lipids, and proteins; this complex structure traps nutrients and minerals from the surrounding environment via a scavenging system [12]. The formation of the surface-located biofilm is complex, and it is commonly completed in the following sequential steps (Figure 2): (a) initial contact with and attachment to the surface, (b) microcolony formation, (c) maturation and formation of an architecture that may be typical for a particular species, and (d) detachment and dispersion of the biofilm. In the first step, microbial cells attach to the surface via their appendages, such as pili and flagella, and other physical factors such as van der Waals forces, electrostatic interactions, and solid–liquid surface tension [13]. The hydrophobicity of the surface may play a role in strengthening the attachment of microbes, as it reduces the force of repulsion between the bacteria and the surface [14,15]. After attachment, the multiplication of the cell population starts leading to the formation of microcolonies [16]. These colonies usually consist of many types of microcommunities that coordinate with one another in multiple aspects such as the exchange of metabolizable substrates and metabolic products. In the subsequent step, maturation and biofilm architecture formation occur. When the microbial cell density reaches a certain threshold, the population regulates community-wide gene expression (a phenomenon known as quorum sensing) via the secretion of signaling molecules known as auto-inducers [17]. These signal molecules play a role in cell multiplication, adhesion, or even detachment, which are important factors in biofilm development. The detachment/dispersion of the biofilm occurs when microbial cells within the biofilm perform quick multiplication and dispersion to convert from sessile to planktonic forms.

### 3.2. Tissue-Located Bacterial Biofilms

Certain bacterial pathogens effectively colonize host tissues and develop biofilm aggregates that lead to infections. Clinical complications resulting from tissue-located biofilms include *P. aeruginosa* infection associated with cystic fibrosis, *S. aureus* involvement in infective endocarditis, and *Helicobacter pylori* contribution to gastric ulcer. A schematic illustration showing *P. aeruginosa* biofilm development within cystic fibrosis airways, as a model of tissue-located biofilm, is depicted in Figure 3. The response of the immune system to initial bacterial infection is primarily innate, involving the activation of macrophages and the complement system, which attracts polymorphonuclear cells (PMNs) to the site of infection [18]. As biofilm formation is initiated and the EPS is produced, PMNs become the dominant immune cell type, with the extent of the immune response depending on the pathogen and the environment. As the acquired immune system is gradually activated, synergy occurs between the innate and acquired immune responses, which accelerates the disease progression by reinforcing PMN migration and enhancing their proteolytic and oxidative activity, resulting in human tissue damage [19]. During biofilm maturation, the bacterial population becomes structurally heterogeneous, consisting of cells with different growth rates and metabolic states (e.g., dormant and persister cells) and bacteria with acquired mutations. In response to tissue-located biofilms, antibodies bind to virulence factors and other bacterial antigens, forming immune complexes. The tolerance of the biofilm to PMNs is mediated by (i) the increased biofilm size which prevents phagocytosis, (ii) the protection provided by the EPS, and (iii) the inactivation of the complement system, opsonization avoidance, and the evasion of immune recognition [20,21]. If the host response reduces the pathogenic biofilm size and causes the dispersion of the biofilm cells, low-grade focal inflammation may occur, and a relapse of infection may follow due to the regrowth of persister cells. Biofilms could form on tissue surfaces, causing inflammation and the destruction of underlying tissues; an example is microbial biofilms on damaged endothelial cells of the heart covered by fibrin and thrombocytes, causing endocarditis [22].

## 4. Role of Key Biofilm Components in Infections

The matrix of bacterial biofilm is made of exopolysaccharides, proteins, eDNA, and lipids. These components collectively contribute to biofilm characteristics, but each add different functions. Due to the crucial involvement of exopolysaccharides and eDNA in biofilm infections and host immune evasion, these two key biofilm components will be emphasized in the current review.

### 4.1. Exopolysaccharides

Exopolysaccharides are important contributors to the biofilm matrix in many organisms including *P. aeruginosa*, nontypeable *Haemophilus influenzae* (NTHI), and *Salmonella* serovars. The role of exopolysaccharides’ individual components has been extensively studied [23,24]. The *P. aeruginosa* genome encodes at least three matrix-related exopolysaccharides, Psl, Pel, and alginate, with each having distinct roles in biofilms formed by mucoid or nonmucoid *P. aeruginosa* strains. NTHI lacks an identifiable exopolysaccharide that contributes to biofilms, but lipooligosaccharide plays a significant role in modulating the biofilm structure [25]. The composition of the *Salmonella* serovars’ biofilm matrix is complex and it varies depending on the environmental conditions and the serovar [26]. In general, exopolysaccharides play a role in promoting resistance to host innate immune components including phagocytes, antimicrobial peptides, and opsonization by complement. The proposed resistance mechanisms include the ability of exopolysaccharides and eDNA to bind to charged antimicrobial peptides, limiting the oxidative capabilities of phagocytic cells, acting as a physical block to surface immunoglobulins, and decreasing complement binding [27]. 

### 4.2. Extracellular DNA

eDNA is derived from random genomic sequences and is considered to be a critical component of the biofilm matrix of several pathogens including *P. aeruginosa* and *Salmonella* serovars. The source of eDNA is the stochastic lysis of a subpopulation of the bacteria within the biofilm, and this eDNA interacts with other components to stabilize the matrix. DNase treatment of biofilms has variable effects on destabilizing the matrix; this variability is likely due to the shielding of eDNA due to its interaction with other biofilm matrix components (e.g., proteins). eDNA provides biofilm cells with resistance by sequestering antimicrobial peptides and aminoglycosides, thus preventing bacterial cell membrane perturbation by these agents [28]. In addition, eDNA binds to human defensins and limits their contact with bacterial cells residing within the biofilm, thus reducing their antimicrobial activity [29]. Moreover, eDNA is a critical component in *Salmonella* biofilm, and provides resistance to antimicrobial peptides via the chelation of Mg^2+^ [28].

## 5. Key Host Innate Immune Responses against Biofilm

Persistent biofilm infections are combated by innate immunity, which is a nonspecific defense mechanism. Nitric oxide and antimicrobial peptides are key molecules that play a substantial role in innate immune responses. These have been found to be particularly effective in fighting *P. aeruginosa* infections associated with cystic fibrosis.

### 5.1. Nitric Oxide

Nitric oxide (NO) is a signaling molecule that has both antimicrobial and immune-regulatory properties in the upper and lower airways. The antimicrobial effect of NO is attributed to the increased ciliary beat frequency, which clears accumulated mucus, and the direct interaction of NO itself with other reactive species such as superoxide yielding peroxynitrite, a highly toxic molecule which oxidizes microbial cellular targets [30,31]. In the nasal and sinus epithelium, NO is produced constitutively in relatively high concentrations, while its production is low in the lower airways. Quorum-sensing molecules produced by bacteria can also induce NO production in the upper airway [32]; however, the extent to which this induction contributes to constitutive NO production is unknown. The NO produced in the upper airway is sufficient to exert antimicrobial effects, as demonstrated by the inhibition of *P. aeruginosa* growth in sinus airway epithelial cells [33]. Overall, NO plays a crucial role in innate immunity in the upper and lower airways, and the impairment of its production may contribute to various biofilm-associated respiratory diseases.

### 5.2. Innate Antimicrobial Peptides

Antimicrobial peptides (AMPs) are essential components of innate immunity. Several AMPs are produced constitutively in respiratory secretions, and their production can be induced via pathogen-sensing receptors. Certain AMPs inhibit biofilm formation. Lactoferrin, the second most abundant AMP in respiratory secretions, blocks biofilm formation via chelating iron; this chelation stimulates twitching motility in bacterial pathogens, which makes it drift across the mucosal surface [34]. The antibiofilm AMP, SPLUNC1/BPIFA1, reduces airway surface tension and inhibits both *P. aeruginosa* and *Klebsiella pneumoniae* biofilm formation [35,36,37]. In patients severely infected with *P. aeruginosa*, the level of SPLUNC1 expression was reduced, and they had a significantly higher need for repeated sinus surgery [38]. SPLUNC1 is unable to function properly under cystic fibrosis conditions despite the importance of this AMP in protection against biofilm-associated infections. These findings suggest that innate AMPs could be one of the first lines of defense against biofilm infections in the human body. 

## 6. Biofilm Eradication Strategies

The seriousness of health risks associated with biofilm-related infections drives the ongoing endeavors to create innovative antibiofilm agents that target extracellular matrix formation, promote biofilm dispersal, or act against the resilient cells within the biofilm core, as illustrated in Figure 4. Approaches to enhance the efficacy of antibiofilm drugs include encapsulating into nanoparticles for optimal delivery or combining multiple drugs to boost antimicrobial activity. However, the cytotoxicity and in vivo treatment efficiency of antibiofilm agents remain crucial concerns. Examples of key current or emerging antibiofilm agents are outlined in Table 1 and further discussed in the subsequent sections.

### 6.1. Targeting Extracellular Polymeric Substances 

#### 6.1.1. Small Molecule Inhibitors

Examples of biofilm-related small molecule inhibitors include those targeting the synthesis of the intracellular signaling molecules (second messengers) often found in bacteria, such as cyclic-di-guanosine monophosphate (c-di-GMP), which regulates EPS-producing enzymes in Gram-positive (e.g., *S. mutans*) and Gram-negative (e.g., *P. aeruginosa*) bacteria. Hence, interrupting the production of c-di-GMP, using small molecule inhibitors, could be implemented as a strategy in combating biofilms and associated infections. Small molecule inhibitors (e.g., catechol-containing sulfonohydrazide compounds) of di-guanylate or di-adenylyl cyclase were identified as potent antibiofilm agents in in vitro biofilm models, but their efficacy against biofilms in vivo requires further validation [39,40]. Another approach involves inhibiting EPS glucan synthesis by glucosyltransferase using small molecule inhibitors (e.g., a quinoxaline derivative), which decreases the accumulation of pathogenic biofilms on teeth and suppresses the onset of oral diseases [41]. An inhibitor targeting the *Candida* interaction with host fibronectin disrupted the biofilm formation by this fungus [42]. Various biomolecules that bind to EPS adhesins, which enable the attachment of biofilm cells to EPSs inside mature biofilms, are being explored as potential treatments for biofilm infections. These approaches offer potential avenues for developing novel therapies to combat the bacterial biofilms associated with persistent infections.

#### 6.1.2. Enzymes Degrading Extracellular Polymeric Substances

Contingent upon their chemical structures, degrading EPSs can be an important strategy in combating bacterial biofilms. Various approaches include the use of exopolysaccharide-degrading enzymes, such as dispersin B, to disrupt the matrix of pathogenic oral biofilms, and glucan hydrolases, which have been used to degrade the EPSs associated with dental biofilms [43,44]. In another approach, purified serine protease, Esp, is used to inhibit *S. aureus* biofilm formation and eradicate pre-existing biofilms *in vitro*, enhancing the susceptibility of biofilm-forming cells to antimicrobial β-defensin 2, and reducing *S. aureus* nasal colonization in humans [45]. 

**Table 1 antibiotics-12-01005-t001:** Summary of selected antibiofilm agents which combat persistent biofilm infections.

Antibiofilm Agent	Target Pathogen	Antibiofilm Mode of Action	Study Model	Reference
Exopolysaccharide-targeting agents				
Quinoxaline derivative	*Streptococcus mutans*	Glucosyltransferase inhibitor	Anticaries rat	[41]
Oxazole derivative	*S. mutans*	Antagonizing glucosyltransferases	Dental caries rat	[46]
Dispersin B	*Staphylococcus spp.*	Inhibited skin colonization, detachment of Staphylococcal cells from skin	In vivo pig model	[47]
Endolysins	*S. aureus*	Peptidoglycan hydrolases	System MRSA infection in mice	[48]
Dornase alfa	*Pseudomonas aeruginosa*	Dissolving cystic fibrosis sputum and fibrillar structures	Cystic fibrosis sputum	[49]
DNABII antibodies	*Haemophilus influenzae*	Targeting epitopes of DNABII found in extracellular DNA	Chinchilla and murine	[50]
α-amylase	*S. aureus* and *P. aeruginosa*	Exopolysaccharide disruption	*Danio rerio*	[51]
*Biofilm dispersion-targeting agents*				
Nitric oxide	*P. aeruginosa*	Biofilm dispersion Reduced biofilm tolerance to antibiotics	Cystic fibrosis sputum	[52]
Cephalosporin-3′-diazeniumdiolates	*P. aeruginosa*	Biofilm dispersion; increases biofilm susceptibility to antibiotics	Microtiter plates	[53]
Nitroxides	*P. aeruginosa*	Promotes biofilm dispersal, inhibits biofilm formation, increases swarming motility	Flow chambers	[54]
Autoinducing peptide inhibitor	*S. aureus*	Quorum sensing inhibitor	RN9222 cell line	[55]
Natural peptide Capsicumicine	*S. epidermidis*	Disassembly of biofilm matrix	SKH1 mice	[56]
Biofilm persister-targeting agents				
TM5 peptide	*P. aeruginosa* and *S. aureus*	Antipersister agent	Laboratory settings	[57]
Rifampin + Fosfomycin	*S. aureus* (Methicillin-resistant)	Cure of cage-associated infections	A foreign body infection model using guinea pigs	[58]
Acyldepsipeptide ADEP4	*S. aureus*	Activation of ClpP protease which kills growing and persister cells	Mouse model of a chronic infection	[59]
Glycosylated cationic peptides	*S. aureus* (Methicillin-resistant)	Bactericidal against persister cells and disperses biofilm mass	Ex vivo wounded human skin infection	[60]

Endolysins, which enzymatically degrade bacterial peptidoglycan, can also be used to target biofilms [48]. DNases can disrupt premature biofilms by degrading eDNA, but this action is likely impacted in mature biofilms by other biomolecules such as exopolysaccharides and proteins, which contribute to biofilm structural integrity [61]. Recombinant human DNase I (dornase alfa) is used therapeutically to break down DNA in sputum from patients with cystic fibrosis [49]. This approach reduces sputum viscosity, improves lung function, and lowers the risk of exacerbation. An intervention study using dornase alfa in patients with early lung disease showed significant improvement, compared to the placebo group, with the potential to use this approach to decrease the rate of lung function decline in children [62]. Combining matrix-degrading enzymes, such as glucano-hydrolases and DNases, with antimicrobial agents enhances biofilm removal and antimicrobial efficacy [63]. In general, the co-administration of EPS synthesis inhibitors or EPS-degrading enzymes, which lack intrinsic antibacterial activity, with antimicrobial agents could serve as a multitarget approach for biofilm removal.

#### 6.1.3. Antibodies and Nucleic-Acid-Binding Proteins

EPS-targeting antibodies and nucleic-acid-binding proteins may be used to combat biofilm infection. Although the use of vaccines faces considerable challenges due to the antigenic variability in biofilm-forming clinical isolates, the use of monoclonal antibodies against specific EPS components has shown promise. Psl-specific antibodies increased the opsonophagocytic killing of *P. aeruginosa*, inhibited pathogens’ adherence to lung epithelial cells, and showed prophylactic protection in several animal models against *P. aeruginosa* infection [64]. Antibodies against the *Enterococcus faecalis* pilus protein (EbpA) prevented EbpA-mediated fibrinogen-dependent bacterial aggregation and biofilm formation on catheters [65]. Polyvalent antibodies that target both planktonic and biofilm-expressed polypeptides from *S. aureus* showed increased antibiofilm efficacy in combination with antibiotics in a rabbit model of osteomyelitis [66]. The DNABII family of DNA-binding proteins provides structural integrity to eDNA. When combined with antibiotic therapy, immunotherapy targeting DNABII has shown efficacy in vivo against biofilms in several bacterial species, including oral bacteria, uropathogenic *E. coli*, and *P. aeruginosa* in a mouse lung infection model [50]. Combined immune and antibiotic therapy has also shown efficacy against methicillin-resistant *S. aureus* (MRSA) biofilms [67]. Furthermore, DNABII antibodies were combined with an integrated host-factor-targeting vaccine, resulting in the disassembly of nontypeable *H. influenzae* biofilms and preventing the associated disease [68]. Such a combinatorial approach for immunization against multiple targets is promising in combating biofilm-associated infections [69].

### 6.2. Biofilm Dispersion-Based Strategies

#### 6.2.1. c-di-GMP Biosynthesis Inhibitors

The biofilm dispersal process represents a new approach to disrupting biofilms. Targeting the metabolic pathway of c-di-GMP, which plays a key role in the biofilm life cycle of both Gram-positive and Gram-negative bacteria, is a plausible strategy, although the complexity of c-di-GMP regulation makes it challenging to control [70]. Nitric oxide has been shown to regulate c-di-GMP accumulation levels and mediate biofilm dispersal. Low-dose gaseous NO (picomolar to nanomolar levels) was shown to reduce the size of *P. aeruginosa* biofilm aggregates in the sputum of patients with cystic fibrosis [52]. Cephalosporin-3′-diazeniumdiolates (C3Ds) are promising biofilm-dispersing drugs, which selectively deliver NO to bacterial biofilms when the bacterial beta-lactamase cleaves the beta-lactam ring and releases NO. C3Ds have shown effectiveness in dispersing *P. aeruginosa* biofilms [53]. Nitroxide analogs are also under development to overcome NO instability and to exert antibiofilm activity in a NO-mimetic fashion [54]. These compounds elicited biofilm dispersal in *P. aeruginosa* and *E. coli* in a similar manner as NO, but failed to disperse MRSA biofilms. Overall, the use of c-di-GMP biosynthesis inhibitors may promote biofilm self-disassembly and make bacteria more susceptible to conventional antibiotics, thereby reducing the likelihood of recolonization.

#### 6.2.2. Quorum Sensing Inhibiting Peptides

Targeting quorum sensing, a strategy that interferes with the cell-to-cell communication systems of bacteria, is a promising approach for the development of novel antibiofilm therapeutics. Quorum sensing inhibitors (QSIs) have been extensively evaluated for their efficacy in clinically relevant bacterial biofilms using in vitro and in vivo models. For example, the development of an autoinducing peptide inhibitor effectively reduced subcutaneous biofilm formation during transplantation in a mouse model [55]. The use of an RNAIII-inhibiting peptide reduced MRSA biofilms in a mouse wound model [71]. Tryptophan-containing peptides interfered with the *Pseudomonas* quorum sensing system and inhibited virulence factor production, biofilm formation, and EPS accumulation [72]. Moreover, the human hormone atrial natriuretic peptide strongly dispersed *P. aeruginosa* biofilms by acting directly on the bacterium AmiC sensor protein and the peptide enhanced the antibiofilm action of multiple antibiotics [73]. However, the action of QSI molecules can be hindered by the biofilm EPS; thus, to be effective, these inhibitors need access to the site of active quorum sensing signaling. In addition, the complexity of cell signaling networks makes it challenging to apply this therapeutic approach, although such inhibitors can be combined with other antibiofilm agents to improve biofilm dispersal efficacy.

### 6.3. Targeting Biofilm Metabolism and Dormancy

#### 6.3.1. Metabolic Inhibitors

The use of certain exogenous amino acids has shown promise in the treatment of biofilms. Amino acids such as L-arginine were found to modulate pH homeostasis and suppress the growth of *Streptococcus mutans* in polymicrobial biofilms [74]. L-methionine has also been identified as a promising adjuvant for treating *P. aeruginosa* biofilms by increasing sensitivity toward ciprofloxacin and degrading eDNA in the EPS [75]. Iron metabolism is critical for biofilm formation by several pathogens. Gallium, which has chemical similarity to iron, inhibits the iron-dependent pathways required for biofilm formation. This ‘Trojan horse’ strategy has been shown to inhibit biofilm formation and reduce bacterial counts in established biofilms in a chronic biofilm lung model [76]. Furthermore, when iron chelators were adjunctively used with tobramycin, they were effective in reducing *P. aeruginosa* in a co-cultured model of human bronchial epithelial cells from a cystic fibrosis patient [77]. Targeting biofilm-related bacterial metabolism is a promising approach for advancing antibiofilm treatment strategies.

#### 6.3.2. Antipersister Peptides

Biofilms pose a challenge for effective antimicrobial treatment, as they provide a protective environment for bacteria, including persister cells that have a key role in drug tolerance. Conventional antimicrobial approaches, which target metabolically active cells, have limitations against dormant or persister cells; therefore, it is valuable to consider novel antipersister agents. Members of AMPs, which are active against slow-growing bacterial pathogens and have broad-acting antimicrobial activity, may be useful against persistent microbial biofilms [78]. A broad-spectrum antimicrobial peptide, TM5, has been recently developed and found to reduce planktonic and persister cells in biofilms formed by both Gram-positive and Gram-negative bacteria, but its in vivo clinical effectiveness needs validation [57]. Antibiotics, which target slow-growing bacteria, such as rifampin and fosfomycin, have exhibited enhanced efficacy, when used in combination, against MRSA biofilms in vivo [58]. The acyldepsipeptide SAAP-148 was highly effective against the persisters of MRSA within a prosthetic joint infection model [79]. Similarly, acyldepsipeptide antibiotic (ADEP4) can activate ClpP protease in dormant persister Gram-positive cells, leading to their death. However, this approach has limitations, as ClpP is not an essential enzyme [59]; therefore, ADEP4 will be ineffective in the case of ClpP mutants. 

Targeting specific bacterial species is also possible using synthetic AMPs that consist of dual functionally-independent moieties. The pore-forming activity of AMPs can target actively respiring cells as well as dormant cells; therefore, the use of these AMPs reduces the potential of developing the corresponding antibiotic resistance. Some peptides also induce the degradation of guanosine pentaphosphate, resulting in the abrogation of biofilms of several bacterial species. Despite these promising findings, there are challenges that may compromise AMPs’ effectiveness, including binding instability to EPS components, sensitivity to microbial proteases, and high costs of synthesis. Although combining AMPs can augment conventional antimicrobial activity, the accessibility of these AMPs to target cells embedded within biofilms, along with their limited stability and durability within the body, remain important issues to address.

## 7. Emerging Antibiofilm Technologies

Current considerable knowledge on biofilm microenvironments and the physiology of biofilm-producing cells, and the recent advancements in fields such as nanoengineering, have enabled the development of innovative, multipronged antibiofilm therapeutic strategies. Nanostructures serve as a versatile platform for creating functionalized nanoparticles designed to specifically target biofilm cells without affecting the host cells or for coating surfaces of medical devices susceptible to biofilm formation. The following sections will address the emerging antibiofilm technologies, including the use of nanostructures in clinical settings to effectively combat biofilms. Examples of such antibiofilm nanostructures are depicted in Table 2.

### 7.1. Antibiofilm Nanoparticles

Nanotechnology is the basis for promising tools to target and treat biofilms. Inorganic nanoparticles, such as silver, can be used as biofilm-targeting agents or as nanocoatings with intrinsic antimicrobial activity. Nanoparticles can function as drug delivery vehicles. Advances in liposomal nanoparticles empower novel approaches for drug delivery. Liposomes are physiologically compatible vesicles, composed of one or more phospholipid bilayers. These liposomes can penetrate biofilms while protecting their load of antimicrobial agents from deleterious interactions with the EPS or enzymatic inactivation and degradation at the infection site by other bacterial or host components [32]. The lipid component of liposomes fuses with the bacterial cell membrane and releases the drug into the cell’s cytosol, thus maximizing the potential of the drug and reducing the cytotoxicity to the host [87]. Components of liposomes such as hydrophobic bile acids (e.g., deoxycholic acid and ursodeoxycholic acid) complexed with the antibiotics kanamycin, vancomycin, and amikacin enhanced *S. aureus* biofilm inhibition, compared to the individual antibiotics [88]. Recent developments in this field include nanoparticles that can be triggered by specific stimuli, such as pH change due to the acidic biofilm environment or enzyme (e.g., external DNase) exposure, to release drugs such as farnesol and ciprofloxacin, or those engineered to selectively target either biofilm matrix constituents or bacteria-specific ligands [80,81]. Nanoparticles can also be used to deliver more than one drug and be functionalized by linking antimicrobial biomolecules on the nanoparticle surface to increase targeting specificity. For example, a nanocomplex was fabricated via encapsulating proteinase K and the photosensitizable Rose Bengal against biofilm infections [82]. Upon exposure to an acidic microenvironment, such a nanocomplex decomposes and releases proteinase K to degrade the proteins in a biofilm matrix, whereas upon illumination, Rose Bengal releases reactive oxygen species to kill bacteria in the biofilm core. Overall, nanoscience enables a promising therapeutic platform and effective biofilm-targeting approaches. However, further advances in this field should focus on enhancing in vivo efficacy and biocompatibility, understanding potential cytotoxicity and the metabolism of nanoparticles in the body, and developing affordable large-scale manufacturing of antibiofilm products for the healthcare market.

### 7.2. Antibiofilm Surface Coatings

Abiotic surfaces may be engineered to inhibit bacterial adhesion and biofilm formation. The incorporation of antibiotics or other biocides into surface coatings has been explored, but the potential deleterious effects of biocides (e.g., silver nanoparticle toxicity in a rat model [89]), the progressive decrease in efficacy over time, and the nonspecific absorption of exogenous surfactants and proteins in these coatings have been considerable hurdles to overcome. Hydrophilic polymers, such as polyethylene glycol, have been used to prevent the biofouling of medical devices. The use of super-hydrophilic or super-hydrophobic surfaces [90,91] has been suggested to decrease bacterial protein deposition and attachment, but their antibiofilm effects are often transient. The development of functionalized medical implant surfaces with a vast range of antimicrobial and antibiofilm coatings such as silver, titanium oxide, and copper has been explored [83]. A self-assembled antimicrobial peptide, GL13K, was decorated with silver nanoparticles and then the complex was used to coat titanium surfaces to combat *Streptococcus gordonii*, the primary colonizer of oral biofilms [84]. This hybrid coating showed promising antimicrobial activity in vivo in a subcutaneous infection rat model. Bottom-to-top approach using nanomaterials as ‘building blocks’, to which antibiotics have been immobilized, may serve as innovative antibacterial surface coatings [92]. 

### 7.3. Antimicrobial Microneedles

Microneedle patches were initially designed for transdermal drug delivery, but have been investigated for a variety of applications, including treating biofilms. Microneedles can penetrate the EPS barrier of biofilms, making this approach an efficient drug delivery system. Researchers have developed microneedle patches consisting of dissolvable microneedle arrays with antibiotic-loaded nanoparticles that can release the antibiofilm drug within the biofilm matrix once exposed to gelatinase produced by resident microbes [85]. This approach seems to be more effective than using free drugs in treating biofilms. Additionally, researchers have developed flexible microneedle array patches that can co-deliver oxygen and antimicrobial agents simultaneously, reducing the bacterial bioburden [93]. A dissolvable microneedle array patch with bacterial-responsive doxycycline-loaded nanoparticles reduced bacterial biofilm by 99.9% in an ex vivo porcine skin biofilm [94]. Researchers have also developed a wound dressing consisting of engineered AMP-loaded dissolvable microneedle arrays that could eliminate MRSA biofilms [95]. This dressing can deliver AMPs precisely to both the interior and exterior parts of biofilms, thus minimizing the recurrence of biofilm-related infections. Conclusively, incorporating multiple antimicrobial agents into water-soluble microneedle arrays holds promise in eradicating difficult-to-treat biofilms.

## 8. Future Directions

During the past two decades, research into microbial biofilms has advanced rapidly, shedding light on the complex nature of this phenomenon. However, the persistence of infections associated with biofilms remains a considerable health crisis. As such, coordinated efforts are needed to deepen our understanding of the genetics, physiology, and dynamics of bacterial biofilms, particularly in relation to chronic infections. Further research should identify the genes responsible for each stage of biofilm development, such as those crucial for the initial transition of individual cells into aggregate forms, as well as the mechanisms by which biofilms develop antimicrobial resistance.

Advancements in transcriptomics, metabolomics, and transposon-based next-generation sequencing could collectively reveal new genetic targets for biofilm research. Meanwhile, the discovery of novel antibiofilm agents that target biofilm-specific bacterial components is necessary. Nanoparticle-based antibiofilm agents represent an emerging field of research that has shown promising results in combating bacterial biofilms, especially when employing multiple antibiofilm agents. Novel small antimicrobial peptides may afford new treatments for biofilm-associated respiratory infections. Understanding the molecular mechanisms by which these peptides modulate biofilm signaling/virulence or affect host immunity will facilitate their potential therapeutic applications.

Biofilms serve as pathogenic niches within the human body, presenting significant challenges to eradication via single-target therapeutic agents that are primarily effective against individual microbial cell type. Combinatorial approaches, which involve the concurrent use of antibiofilm drugs with different modes of action and biofilm-targeted immunotherapy, may help to (i) simultaneously degrade EPSs, induce biofilm dispersal, and eliminate persister cells, thus significantly increasing the potential for the successful clinical eradication of established biofilms, and (ii) overcome antimicrobial resistance that arises from the use of single antibiotics. Additionally, the development of safe and on-demand antibiofilm drug delivery systems is crucial to avoid excessive drug dosages that could increase cytotoxicity to the host and antibiotic resistance to the biofilm microbes. Drug delivery systems that specifically target bacterial components (e.g., lipopolysaccharides) or bacterial metabolites (e.g., endotoxins) can optimize drug effectiveness and specificity to the biofilm microenvironment. Despite many recent advancements, currently used and proposed antibiofilm agents, and their in vivo delivery, require extensive research to ensure their effective and safe application in clinical settings. The translation of multifaceted antibiofilm drugs from controlled in vitro or in vivo-like environments into real-world clinical settings requires a concerted effort across multiple disciplines. Biomedical researchers, microbiologists, chemists, and engineers all have pivotal roles to play in propelling this crucial initiative forward, bringing the healthcare discipline closer to more effective solutions for the management of biofilm-associated infections. Moreover, industry stakeholders may take the lead on upscaling the production of antibiofilm formulations, thereby facilitating their wider clinical testing and use.

The economic burden of biofilm infections is significant, reflecting the high risks of the associated human diseases. A significant proportion of microbial infections in the human body can be traced back to biofilms [96]. These biofilms exhibit considerable resistance to conventional antimicrobial interventions, often necessitating surgical procedures for effective resolution. The existence of persister cells within the biofilm core further complicates the treatment strategies. These persisters demonstrate a high level of tolerance to antimicrobials, rendering chemotherapy often ineffective, and leading to an increased reliance on surgical intervention. Thus, the financial burden associated with treating persistent biofilm infections is notably high, urging the need for advanced research and innovative therapeutic approaches in this critical area.

## Figures and Tables

**Figure 1 antibiotics-12-01005-f001:**
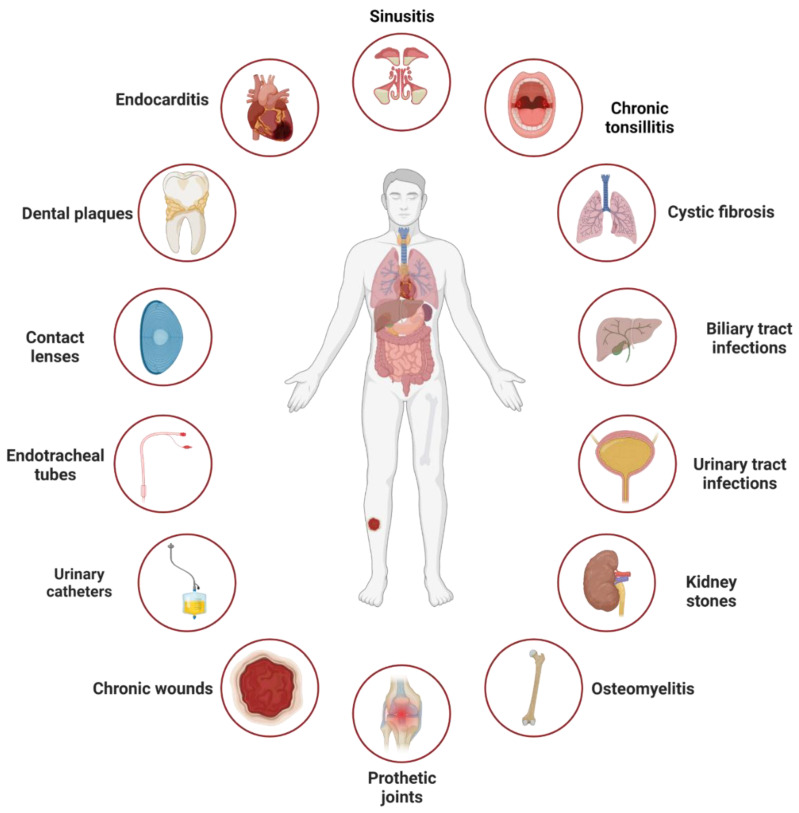
Infections linked to surface- and tissue-located biofilms. These include infections resulting from biofilm formation on surfaces such as catheters, teeth, and kidney stones, or on tissues such as cystic fibrosis lungs, chronic wounds, and tonsils. The figure was adapted from [11] and created using biorender.com, accessed on 28 May 2023.

**Figure 2 antibiotics-12-01005-f002:**
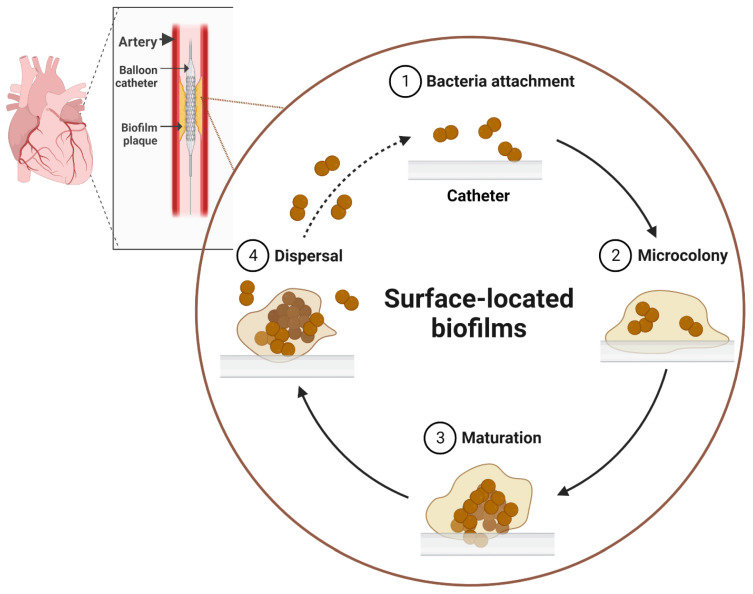
Development of biofilm on abiotic surfaces triggering human infections. Medical devices, such as catheters, are ideal abiotic surfaces for biofilm formation. The process begins with individual cells attaching to the device surface, followed by the secretion of extracellular polymeric substances (EPSs) and the aggregation of biofilm cells (microcolony formation). Subsequently, the full maturation of the biofilm structure occurs via an increased production of EPSs and a rise in biofilm population density. Eventually, biofilm dispersal takes place, causing recurring infections by restarting the biofilm development process. The figure was created using biorender.com, accessed on 28 May 2023.

**Figure 3 antibiotics-12-01005-f003:**
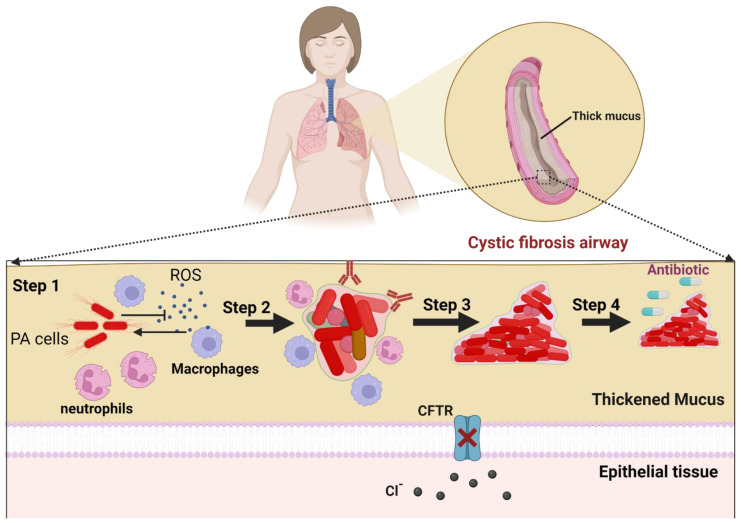
Biofilm-associated infections within cystic fibrosis airways. The biofilm development is a multistep process starting with planktonic *P. aeruginosa* cells inhibiting the host immune responses including neutrophiles and macrophages (step 1) within the thickened mucus accumulated in cystic fibrosis airways (CFTR deficient). Bacterial survivors lose motility, accumulate extracellular polymeric substances, and form biofilm aggregates with heterogeneous populations (step 2). The biofilm populations exhibit genotypic and phenotypic convergence (step 3) to yield a fully mature biofilm with tolerance to antibiotics and persistent populations, which can cause recurring infections (step 4). CFTR, cystic fibrosis transmembrane conductance regulator; ROS, reactive oxygen species; PA, *P. aeruginosa*. The figure was created using biorender.com, accessed on 28 May 2023.

**Figure 4 antibiotics-12-01005-f004:**
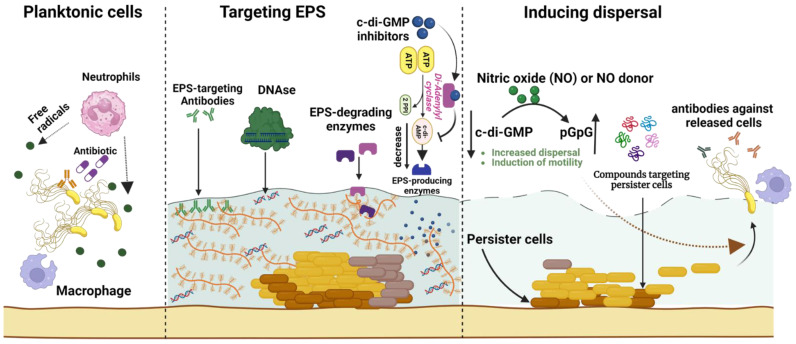
A comprehensive overview of biofilm eradication strategies for overcoming infection scenarios. These strategies include targeting biofilm extracellular polymeric substances (EPSs) via degradation enzymes (e.g., DNases or glucanohydrolases), EPS-specific antibodies, and cyclic-di-guanosine monophosphate (c-di-GMP) inhibitors that subsequently reduce EPS production. Additional approaches facilitate biofilm dispersal by employing nitric oxide to activate proteins that hydrolyze c-di-GMP, thereby boosting biofilm cell dispersal and motility. These dispersed cells become susceptible to destruction by innate immune cells (such as macrophages and neutrophils) via phagocytosis and free radical release, or due to antibiotic treatment. The diagram also depicts antibiofilm agents (e.g., small peptides) that specifically target persister cells within the biofilm core to eradicate recurring biofilm-associated infections. The figure was created using biorender.com, accessed on 28 May 2023.

**Table 2 antibiotics-12-01005-t002:** Efficacy of selected antibiofilm nanostructures designed combat persistent biofilm infections.

Antibiofilm Agent	Target Pathogen	Antibiofilm Mode of Action	Study Model	Reference
Farnesol-loaded nanoparticles	*Streptococcus mutans*	Attenuated biofilm virulence	Dental caries disease model	[80]
Ciprofloxacin-loaded nanoparticles	*Pseudomonas aeruginosa*	Prevented biofilm formation and reduced established biofilm mass	Macrophages	[81]
Proteinase K and Rose-Bengal-loaded nanocomplex	*Staphylococcus aureus*	Biofilm eradication	Cutaneous wound infection in mouse model	[82]
Nanostructured silver antibacterial surfaces	*S. aureus* and *P. aeruginosa*	Antibacterial and antifouling activity	Polydimethylsiloxane films	[83]
AMP * nanostructures with silver nanoparticles	*S. aureus (Methicillin-* *resistant)*	In vivo antimicrobial activity	Subcutaneous infection model in rats	[84]
Microneedle patches with chloramphenicol-loaded nanoparticles	*Vibrio vulnificus*	Biofilm disruption and antibiotic penetration	In vitro biofilm model	[85]
BNN6 ^†^-loaded polydopamine nanoparticles	*S. aureus* *(Methicillin-resistant)*	Decrease in biofilm cells and wound healing	In vivo wound in mouse model	[86]

^†^ BNN6: N,N’-di-sec-butyl-N,N’-dinitroso-p-phenylenediamine; * AMP: Antimicrobial peptide.

## Data Availability

All data are included within the manuscript.

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
