# Peer review of "Combating Bacterial Biofilms: Current and Emerging Antibiofilm Strategies for Treating Persistent Infections"

_antibiotics, 2023, doi:10.3390/antibiotics12061005_

Round 1

Reviewer 1 Report

Title: Combating Bacterial Biofilms: Current and Emerging Antibiofilm Strategies for Treating Persistent Infections

This manuscript is well-written by the authors. I do believe that if they can improve the manuscripts following all comments. It might have a chance to publish in the journal.

Comments

1. Line 22; please re-write using passive voice.

2. Keywords: the maximum keywords are 5 or 6

3. Please include objectives at the end of introduction

4. Introduction; the novelty and original research were not well defined in this introduction

5. Introduction; What distinguishes your research between other? please give state of the art of your research

6. Line 79; there are 2 full stops, please delete.

7. Line 78; please describe more detail on the figure.

8. Line 109; please replace S. aureus instead of Staphylococcus aureus.

9. Line 143 and 150; please replace P. aeruginosa instead of Pseudomonas aeruginosa

10. Table 1, Line 91; please check the font of the alphabet.

11. Please include conclusion or summary in the end of the manuscript.

12. Please delete some unnecessary references or old references

This manuscript is well-written by the authors.

Author Response

Reviewer 1:

This manuscript is well-written by the authors. I do believe that if they can improve the manuscripts following all comments. It might have a chance to publish in the journal. 

  1. Line 22; please re-write using passive voice.

Author response: Edited as suggested.

  1. Keywords: the maximum keywords are 5 or 6

Author response: We removed two of the keywords considering that the Journal allows three to ten keywords.

  1. Please include objectives at the end of introduction

Author response: We have added a paragraph outlining the objectives, original thoughts, and a state of the art of this review.

  1. Introduction; the novelty and original research were not well defined in this introduction

Author response: This has been emphasized in the paragraph already added per the reviewer’s previous comment.

  1. Introduction; What distinguishes your research between other? please give state of the art of your research

Author response: Edited as suggested. Please see previous comments. Additionally, information has been added to Section 8 in the revised manuscript.

  1. Line 79; there are 2 full stops, please delete.

Author response: Edited as suggested.

  1. Line 78; please describe more detail on the figure.

Author response: Edited as suggested.

  1. Line 109; please replace S. aureusinstead of Staphylococcus aureus.

Author response: Edited as suggested.

  1. Line 143 and 150; please replace P. aeruginosainstead of Pseudomonas aeruginosa

Author response: Edited as suggested.

  1. Table 1, Line 91; please check the font of the alphabet.

Author response: We checked the font in Table 1 to match the Journal requirements.

  1. Please include conclusion or summary in the end of the manuscript.

Author response: We have originally integrated our conclusion into the last section (Section 8. Future directions). To avoid redundancy, we have amended “Section 8: Future directions” with more concluding statements.

  1. Please delete some unnecessary references or old references

Author response: We attempted to minimize the use of references as we could; however, we have found all cited references, including the few old ones, are needed to support the scientific statements we made throughout the manuscript. However, we have added two recent references (published in 2022): Reference # 95 and 96.

Reviewer 2 Report

  1. The paper titled “Combating Bacterial Biofilms: Current and Emerging Antibiofilm Strategies for Treating Persistent Infections highlighted the clinical problems associated with biofilm infections and focused on current and emerging antibiofilm strategies. The manuscript is written well and has potential but need following changes before consideration
  2. Title is fine and describing the manuscript in elaborative way
  3. Abstract is not written good as lack manuscript background and application of study literature review so add them with focus on the stakeholders for which this review findings are useful at the end of abstract  
  4. Overall, the review has been divided in several headings with detailed description for the causes, potential impact on human health as well as the antibiofilm strategies for the control of biofilm. Although there is very useful review of literature but it lack economic burden of biofilm formation and its economic impact and addition of this information will add a value in the manuscript.  
  5. Do not use abbreviations in headings like “EPS-targeting strategies” and follow this for all
  6. There is only one table “Table 1. Summary of selected antibiofilm agents which combat persistent biofilm infections” and there is a need for 1 at least and 2 preferably new tables for data presentation in the manuscript
  7. In conclusion major focus should be on findings with practical application as well as make it concise
  8.  
  1. I have checked the manuscripit and grammatical mistakes observed on few places so there is need to go through the paper for language and grammatical mistakes

Author Response

Response to Reviewers

Reviewer 2

The paper titled “Combating Bacterial Biofilms: Current and Emerging Antibiofilm Strategies for Treating Persistent Infections” highlighted the clinical problems associated with biofilm infections and focused on current and emerging antibiofilm strategies. The manuscript is written well and has potential but need following changes before consideration.

  1. Title is fine and describing the manuscript in elaborative way.

Author response: We thank the reviewer for the comment.

  1. Abstract is not written good as lack manuscript background and application of study literature review so add them with focus on the stakeholders for which this review findings are useful at the end of abstract.

Author response: We have added new statements, addressing the points raised by the reviewer, at the end of the abstract.

  1. Overall, the review has been divided in several headings with detailed description for the causes, potential impact on human health as well as the antibiofilm strategies for the control of biofilm. Although there is very useful review of literature but it lack economic burden of biofilm formation and its economic impact and addition of this information will add a value in the manuscript. 

Author response: We have cited a reference (#96) referring to the economic burden associated with biofilm infections. This information is now included in “Section 8: Future directions.”

  1. Do not use abbreviations in headings like “EPS-targeting strategies” and follow this for all

Author response: Edited as suggested.

  1. There is only one table “Table 1. Summary of selected antibiofilm agents which combat persistent biofilm infections” and there is a need for 1 at least and 2 preferably new tables for data presentation in the manuscript.

Author response: As suggested by the reviewer, we have created a new Table (Table 2) that improves the manuscripts and adds depth to it.

  1. In conclusion major focus should be on findings with practical application as well as make it concise

Author response: We have emphasized the practical applications in the new information added to the abstract and to section 8. In this review, we aimed at outlining the pros and cons of each antibiofilm technology. However, we also addressed in more details the challenges facing each antibiofilm technology, which hinder its practical applications.

Comments on the Quality of English Language

  1. I have checked the manuscript and grammatical mistakes observed on few places so there is need to go through the paper for language and grammatical mistakes.

Author response: We have proof-read the manuscript and corrected few grammatical errors. 

Reviewer 3 Report

The manuscript provided a review of bacterial biofilm, including an overview of its role in infection, its major components, and its interactions with the immune system. The authors further provide a review of the biofilm eradication strategies and prospects of antibiofilm technologies. Overall, the manuscript provided up-to-date information in the field. One major concern is the reduction of redundancy / repeated information in the review.

1.       Comment for authors and/or the editors: Please rearrange the figure and text so that they are close for readers to refer to. For example, Figure 2 is related to section 3.1 while separated in the manuscript.

2.       For section 4, you also mentioned proteins and lipids were components of the biofilm. Please consider adding reviews on those or explain why they were not included.

3.       Line 212: CF conditions. Please add the full name for the acronym. Please consider adding a summary sentence to the section.

4.       Please also check the use of acronyms throughout the manuscript making sure that they all have proper full names when they first appear—for example, line 359.

5.       Table 1: please consider making the titles of the three categories more visible.

6.       Please highlight the new information from this review that differentiates from previous ones. 

Please refer to the suggestions. 

Author Response

Reviewer 3:

Comments and Suggestions for Authors

The manuscript provided a review of bacterial biofilm, including an overview of its role in infection, its major components, and its interactions with the immune system. The authors further provide a review of the biofilm eradication strategies and prospects of antibiofilm technologies. Overall, the manuscript provided up-to-date information in the field. One major concern is the reduction of redundancy / repeated information in the review.

Author response: We thank the reviewer for the feedback. In the revised manuscript, we have avoided redundancies as much as possible. Please not that it may appear that a certain topic is revisited in different sections, but each occurrence displays a distinct perspective on the issue at hand. For instance, when discussing antimicrobial peptides, we explore them from varying angles - as a component of the innate immune response, as inhibitors of quorum sensing, and as a strategy to combat persister cells. Each of these discussions addressed the same topic but from a different angle.

  1. Comment for authors and/or the editors: Please rearrange the figure and text so that they are close for readers to refer to. For example, Figure 2 is related to section 3.1 while separated in the manuscript.

Author response: We have moved Figure 2 into the appropriate location as suggested.

  1. For section 4, you also mentioned proteins and lipids were components of the biofilm. Please consider adding reviews on those or explain why they were not included.

Author response: We have included exopolysaccharides and eDNA because of the presence of significant scientific evidence supporting their role in biofilms infections and immune evasion. We added a statement in the revised manuscript saying, “Due to the crucial involvement of exopolysaccharides and eDNA in biofilm infections and host immune evasion, the current review will primarily focus on these two key components.

  1. Line 212: CF conditions. Please add the full name for the acronym. Please consider adding a summary sentence to the section.

Author response: Changes were made as suggested by the reviewer.

We have added a statement saying, “Thus, innate AMPs could be one of the first lines of defense against biofilm infections in the human body.”

  1. Please also check the use of acronyms throughout the manuscript making sure that they all have proper full names when they first appear—for example, line 359.

Author response: We have checked abbreviations as suggested. Regrading “SAAP-148”, it is not an abbreviation but is the name reported by the authors for the acyldepsipeptide investigated.

  1. Table 1: please consider making the titles of the three categories more visible.

Author response: Edited as suggested.

  1. Please highlight the new information from this review that differentiates from previous ones.

Author response: We have modified the introduction and the final section (Future directions) to emphasize the distinction of this article from the previously published ones.

Reviewer 4 Report

As a Review paper,  it covers all the areas related to the title. 

Author Response

We thank the reviewer for the feedback and for taking the time to review the manuscript.

Reviewer 5 Report

The manuscript presents a comprehensive literature review on biofilms and their eradication strategies. The roles and characteristics of biofilms in persistent infections are described in detail from beginning to end. Current and emerging biofilm eradication strategies are organized by their mode of action and discussed. However, there are already numerous reviews on biofilms and their eradication strategies. And while this review is well written, it does not provide sufficient new information. I suggest the authors  give more in depth insights focused on a specific niche to differentiate their review from others. For example, in section 6 and 7,  further discussion on biofilm eradication strategies in terms of overlooked interactions or unintended consequences on  “persistent infections” and the patient’s immune system. 

Some other minor comments are below:

I recommend the authors bold the figure titles to distinguish them from the figure caption. 

Figure 4. On pg 8 could be simplified a bit to make it easier to read. In particular, the c-di-GMP inhibitors figure is a bit small and complex. The multiple icons depicting EPS and extracellular DNA could also be reduced by a bit. 

In section 7.1 on pg12, the authors mention enzyme exposure and pH change or acidic microenvironments as release triggers for nanoparticles. For clarity, the authors should briefly state that these stimuli are present in biofilms and include a citation.

Author Response

Reviewer 5:

The manuscript presents a comprehensive literature review on biofilms and their eradication strategies. The roles and characteristics of biofilms in persistent infections are described in detail from beginning to end. Current and emerging biofilm eradication strategies are organized by their mode of action and discussed. However, there are already numerous reviews on biofilms and their eradication strategies. And while this review is well written, it does not provide sufficient new information. I suggest the authors give more in depth insights focused on a specific niche to differentiate their review from others. For example, in section 6 and 7, further discussion on biofilm eradication strategies in terms of overlooked interactions or unintended consequences on “persistent infections” and the patient’s immune system. 

Author response: We thank the reviewer for the feedback. The revised manuscript includes the following modifications that are in line with reviewer’s recommendations:

  • We have added a paragraph at the end of the “Introduction” to underscore the contemporary relevance and unique focus of this review. Particularly, we emphasize the promising potential of small molecules as potent agents in the battle against biofilm infections.
  • We have added information in Section 8 to highlight the economic burden of biofilm infections.
  • We have added more information (Section 8) emphasizing the great promise of multitargeted antibiofilm strategies in the management of biofilm-associated infections.
  • Sections 6 and 8 discusses the strategies for eradicating biofilms, particularly those situated in specific niches in infected tissues or on the surfaces of medical devices employed in clinical settings. This focus is mirrored in Table 1 and the new Table 2.
  • “Section 3.2. Tissue-located bacterial biofilms” now elaborates on the immune responses to biofilm infections.

Some other minor comments are below:

I recommend the authors bold the figure titles to distinguish them from the figure caption. 

Author response: Edited as suggested.

Figure 4. On pg 8 could be simplified a bit to make it easier to read. In particular, the c-di-GMP inhibitors figure is a bit small and complex. The multiple icons depicting EPS and extracellular DNA could also be reduced by a bit. 

Author response: Figure 4 has been modified. We reduced icons depicting EPS and eDNA as suggested and enlarged c-di-GMP inhibitors for increasing readability.  

In section 7.1 on pg12, the authors mention enzyme exposure and pH change or acidic microenvironments as release triggers for nanoparticles. For clarity, the authors should briefly state that these stimuli are present in biofilms and include a citation.

Author response: The statement has been modified to read: “Recent developments in this field include nanoparticles that can be triggered by specific stimuli such as pH change due to the acidic biofilm environment or enzyme (e.g., external DNase) exposure to release drugs such as farnesol and ciprofloxacin…”. We feel the cited references for this statement are adequate.

Round 2

Reviewer 3 Report

Thanks for the response. 

No further comment.